# Low dose ionizing radiation strongly stimulates insertional mutagenesis in a γH2AX dependent manner

Alex N. Zelensky[1,2]*, Mascha Schoonakker[1], Inger Brandsma[1], Marcel Tijsterman[3], Dik C. van Gent[1,2], Jeroen Essers[1,4,5], Roland Kanaar[1,2]*

**1** Department of Molecular Genetics, Erasmus University Medical Center, Rotterdam, The Netherlands, **2** Oncode Institute, Erasmus University Medical Center, Rotterdam, The Netherlands, **3** Human Genetics, Leiden University Medical Center, Leiden, The Netherlands, **4** Department of Radiation Oncology, Erasmus University Medical Center, Rotterdam, The Netherlands, **5** Department of Vascular Surgery, Erasmus University Medical Center, Rotterdam, The Netherlands

* a.zelensky@erasmusmc.nl (ANZ); r.kanaar@erasmusmc.nl (RK)

**Data Availability Statement:** All relevant data are within the manuscript and its Supporting Information files

## Abstract

Extrachromosomal DNA can integrate into the genome with no sequence specificity producing an insertional mutation. This process, which is referred to as random integration (RI), requires a double stranded break (DSB) in the genome. Inducing DSBs by various means, including ionizing radiation, increases the frequency of integration. Here we report that non-lethal physiologically relevant doses of ionizing radiation (10–100 mGy), within the range produced by medical imaging equipment, stimulate RI of transfected and viral episomal DNA in human and mouse cells with an extremely high efficiency. Genetic analysis of the stimulated RI (S-RI) revealed that it is distinct from the background RI, requires histone H2AX S139 phosphorylation (γH2AX) and is not reduced by DNA polymerase θ (*Polq*) inactivation. S-RI efficiency was unaffected by the main DSB repair pathway (homologous recombination and non-homologous end joining) disruptions, but double deficiency in MDC1 and 53BP1 phenocopies γH2AX inactivation. The robust responsiveness of S-RI to physiological amounts of DSBs can be exploited for extremely sensitive, macroscopic and direct detection of DSB-induced mutations, and warrants further exploration *in vivo* to determine if the phenomenon has implications for radiation risk assessment.

## Author summary

Not all DNA in mammalian nuclei is organized into chromosomes. The pool of extrachromosomal DNA molecules is produced by natural process: during genomic DNA repair, viral infections, phagocytosis; and in experimental settings after transfection, and in gene therapy. Extrachromosomal DNA can integrate into the genome at the site of a double-stranded DNA break (DSB), producing a mutation in the chromosome. Because DSBs are dangerous lesions, they are actively eliminated, and their availability is a limiting factor for extrachromosomal DNA integration. It has long been known that inducing additional random DSBs, for example by exposing cells to ionizing radiation, can increase the

**Funding:** RK was funded by Gravitation program CancerGenomiCs.nl from the Netherlands Organization for Scientific Research (NWO, nwo.nl) RK funded by Oncode Institute (oncode.nl), which is partly financed by the Dutch Cancer Society (kwf.nl). DvG, JE were funded by European Atomic Energy Community's Seventh Framework Programme (FP7/2007-2011) under grant agreement n° 249689. The funders had no role in study design, data collection and analysis, decision to publish, or preparation of the manuscript.

**Competing interests:** The authors have declared that no competing interests exist.

frequency of random integration (RI), however the irradiation doses that were used were non-physiological. We found that much smaller doses of radiation, in the upper range of radiological diagnostic procedures, stimulate integration even more efficiently than the much larger doses studied previously. The potency of stimulation is remarkable, given that biological effects of such low doses are generally difficult to register, and warrants re-evaluation using animal models. Surprisingly, the genetic dependencies of radiation-stimulated integration do not include DSB repair pathways, and are distinct from background integration events. Our observations provide a hyper-sensitive tool to detect mutagenesis and reveal new information about the genetic interactions between DNA damage signaling and repair system components.

## Introduction

Extrachromosomal DNA (ecDNA)–endogenous, viral or transfected–can integrate into the genomic DNA, resulting in an insertional mutation. This type of mutagenesis has been primarily studied in the context of exogenous DNA that enters the nucleus as a result of transfection or viral infection, and has several important practical implications [1]. It is used to produce transgenic cell lines and organisms for research and biotechnological applications. Random integration (RI) of transcription-blocking constructs has been exploited as a form of untargeted but traceable mutagenesis ("gene trapping"). Integration of exogenous DNA is an important factor in several therapeutic approaches, where it is regarded as beneficial (stable restoration of a missing gene) or dangerous (insertion near an oncogene and its activation). Viral integration into the genome has been considered as a contributing factor in oncogenesis, even for viruses that do not encode an active integration function [2]. During precise homology-driven modification of the genome (gene targeting), RI of the targeting construct is an unwanted side effect that severely limits the application of this powerful technique in the vast majority of organisms [3].

The presence of extrachromosomal DNA is a physiological condition, as a sizable pool of it exists in the majority of normal cells in tissues, and includes fragments of nuclear and mitochondrial DNA released due to damage repair, telomeric DNA circles [4], non-integrating viral genomes [5], mobile genetic elements and phagocytized extracellular DNA [6]. According to one estimate, the relative fraction of such extrachromosomal DNA in normal tissues can be substantial, reaching 0.1–0.2% of total DNA content [7], which is comparable to other major genomic components such as telomeres (0.4%). How episomal DNA interacts with the genomic DNA and repair systems is not well understood.

Insertion of exogenous DNA into a chromosome can be described by a simple and intuitive model as mis-repair of a double strand break (DSB) in the genomic DNA by non-homologous end joining that traps an extrachromosomal DNA fragment that happened to be in the vicinity of the DSB [8,9]. This model predicts that the proximity of an ongoing DSB repair event to an extrachromosomal DNA molecule will determine the frequency of the insertion, and therefore that increasing the frequency of DSBs above background by inflicting additional damage will increase the likelihood of integration. This prediction has been confirmed numerous times in various cell lines and with different DNA vectors using doses of > 0.5 Gy of ionizing radiation (γ- and X-rays), which is arguably the best-studied method of DSB induction [10–23]. RI stimulation by DSB-inducing chemicals and enzymes has also been demonstrated, as well as by some non-DSB inducing genotoxic agents [16,21,24–33]. In the latter case the stimulation can be explained by indirect DSB induction during replication.

Genomic integration of extrachromosomal DNA is referred to in the existing literature as RI, illegitimate recombination, illegitimate integration, stable integration, stable transformation, non-homologous integration or insertional mutation [8]. Although it is not perfect, we chose to use the first term. In the context of RI stimulated by DNA lesions we use the term stimulated RI (S-RI). Here we report the following, and to our knowledge so far unidentified, properties of the S-RI phenomenon. Firstly, we found that extremely low doses of ionizing radiation (10–50 mGy), similar to those encountered during routine medical diagnostic procedures, strongly stimulate the integration of transfected DNA and episomal viral DNA. Secondly, a screen of multiple knock-out mouse embryonic stem (ES) cell lines revealed that contrary to expectation, disruption of the two major DSB repair pathways has no effect on S-RI. And thirdly, we showed that phosphorylation of H2AX on serine 139 and recruitment of the adaptor protein MDC1, involved in DNA damage response, are essential for the process.

## Results

### Integration is strongly stimulated by physiologically relevant doses of radiation

To investigate the effect of low dose irradiation (< 1 Gy) on RI we transfected ES cells by electroporation with circular or linearized plasmid DNA containing a puromycin resistance gene, divided the cells equally over several culture dishes, and irradiated the dishes using $^{137}$Cs γ-irradiation source with a set of doses ranging from 0.01 to 1 Gy (Fig 1A and 1B). Remarkably, even the lowest dose tested already led to an increase in the number of puromycin-resistant colonies formed after 6–8 days of selection (Fig 1B), with a 7.5±0.8 -fold increase at a dose of 200 mGy. The response was linear between 10 and 200 mGy, then plateaued between 200 and 500 mGy, and decreased at 1 Gy.

The sensitivity of the response to the extremely low doses, the plateauing dose response, and the high magnitude of the stimulation (up to 10-fold) distinguish our findings from numerous previous reports of the phenomenon, as they generally studied doses above 1 Gy [12–19,22]. It is remarkable that the exquisite sensitivity of the assay to physiologically relevant amounts of induced DNA damage appears to have escaped experimental scrutiny for more than half a century. Intrigued by this, we went on to verify the generality of the S-RI phenomenon using different DSB induction methods, cell lines, DNA vectors and assay endpoints. The lower end of the dose range we tested overlaps the dose range of certain medical imaging procedures (e.g. 1–30 mGy for computed tomography (CT) and fluoroscopy [34]). We scanned freshly transfected ES cell suspensions in a micro-CT instrument (Quantum FX, PerkinElmer) used for mouse imaging. One to five sequential scans were performed at the lowest resolution, each scan delivering 13 mGy based on the manufacturer's data. A single scan produced a clear increase in integration frequency (2.28 ±0.17, n = 7) and consecutive scans resulted in a dose response curve similar to what we observed with the $^{137}$Cs source (Fig 1C). Human HeLa cells transfected with the plasmid DNA by electroporation or lipofection performed similarly to mES cells in the S-RI colony formation assay (S1A and S1B Fig).

To confirm that our observations were not limited to plasmid DNA transfection or antibiotic selection, we performed S-RI experiments using non-replicating episomal viral vectors and with fluorescent marker detection by FACS. Recombinant adeno-associated virus type 2 (rAAV2) has a single-stranded DNA genome, and while it can integrate in cultured human cells with some sequence specificity conferred by the *rep* protein, wild-type AAV2 infecting human tissues and *rep*-deleted rAAV2 vectors persist as episomes and integrate infrequently and with no detectable specificity [5,35]. We added lysates containing rAAV2 particles encoding GFP to the HeLa cell culture media, incubated overnight to allow infection to occur, re-

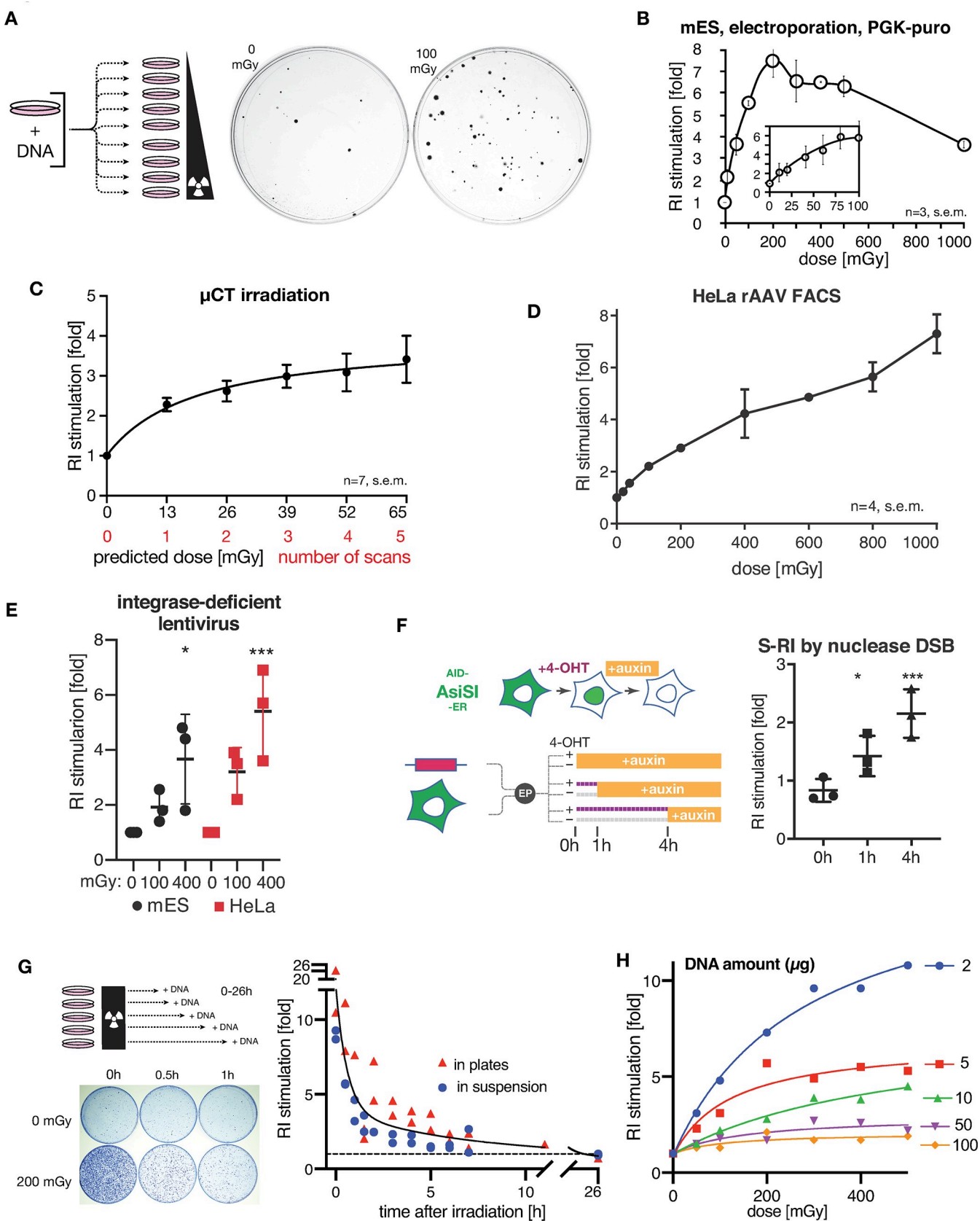

**Fig 1. RI is strongly stimulated by low doses of ionizing radiation.** (**A**) Scheme of the typical S-RI assay, in which antibiotic-resistant colony formation is used as an end point. Cells are transfected with linear or circular plasmid DNA, divided over a series of dishes and irradiated with different doses. After 6–8 days of antibiotic selection, plates are stained and colonies counted. Representative examples of an unirradiated plate and a plate irradiated with 100 mGy are shown at the right. (**B**) Mouse ES cells were electroporated with linearized plasmid with puromycin resistance gene. Colony numbers were normalized to unirradiated control to give the fold increase in RI efficiency, which is plotted. (**C**) mES cells were electroporated with linear or circular pLPL plasmid, divided evenly into 6 vessels, and subjected to the indicated number of low-resolution micro-CT scans in a Quantum FX instrument. Indicated doses are based on manufacturer's specification. n = 7, error bars show s.e.m. (**D**) S-RI in HeLa cells infected with rAAV2-GFP and irradiated with the indicated doses one day later. Cells were maintained without selection, passaged when they reached confluency and sampled by FACS up to day 14. Representative FACS plots are shown in (S1C Fig). The fraction of GFP positive cells in irradiated cultures normalized to unirradiated control is plotted. (**E**) S-RI in mES and HeLa cells infected with IDLV carrying puromycin-resistance gene and irradiated with the indicated doses. Ratio of colony numbers in irradiated to unirradiated plates determined in each experiment is plotted. Statistical significance indicated by asterisks was determined by ANOVA: * $p \leq 0.05$, *** $p \leq 0.001$. (**F**) S-RI by nuclease DSB. AID-AsiSI-ER U2OS cells containing a stably integrated coding sequence for AsiSI nuclease tagged with auxin-inducible degron (AID) and estrogen receptor (ER) domain that triggers re-localization from cytoplasm to the nucleus upon addition of tamoxifen (4-OHT). Cells were electroporated with linearized plasmid PGK-puro DNA and seeded immediately into dishes containing either 0 or 300 nM 4-OHT. Auxin was added at the indicated time points to induce AsiSI degradation. The ratio of puromycin resistant colony numbers formed in plates that contained 300 nM 4-OHT was divided by the number of colonies in 0 nM plates. Statistical significance indicated by asterisks was determined by ANOVA: * $p \leq 0.05$, *** $p \leq 0.001$. (**G**) RI stimulation by irradiation of mES cells before transfection with linearized PGK-puro plasmid. Cells were irradiated either in plates (as shown on the scheme on the left) or in suspension. Data from four independent experiments is plotted. (**H**) Effect of the amount of transfected DNA on S-RI efficiency determined using colony-formation S-RI assay.

seeded the cells into a series of dishes and irradiated them with 0.02–1 Gy. The fraction of cells expressing GFP was ~60% two days after the infection and gradually decreased over time due to dilution and loss of rAAV2-GFP episomes in dividing cells. At day 13 only 0.05–0.1% of unirradiated cells remained GFP-positive, presumably due to stable integration [36]. The GFP-positive fraction increased linearly with the dose in irradiated cells (Fig 1D, S1C Fig). Transfection of human U2OS cells with plasmid DNA containing a GFP minigene, followed by FACS 14–21 days later, further confirmed that selection is not required to observe the S-RI effect (S1D Fig). We also observed S-RI (Fig 1E) with integrase-deficient HIV-1 lentiviral (IDLV) vectors [37,38] whose RNA genomes are reverse-transcribed normally, but accumulate as circular or linear episomal DNA because of the inactivating D64V mutation in the integrase [39].

Both IDLV and rAAV vectors were used previously to detect nuclease-induced DSBs [36,40,41]. To test whether transfected plasmid DNA also integrates at nuclease-induced DSBs we used the DiVA-AID cell line, which allows precise control over nuclear localization and degradation of the AsiSI nuclease that has hundreds of recognition sites in human genome [42]. We added 4-hydroxytamoxifen to induce AsiSI breaks immediately after the transfection of plasmid DNA and controlled the DSB dose by inducing AsiSI degradation by adding auxin to the media 1 or 4 hours later (Fig 1F). The dose-dependent increase in the number of colonies we observed was consistent with the results we obtained using ionizing radiation.

To better characterize the parameters affecting S-RI we studied its dependence on the timing of irradiation and on the amount of transfected DNA. The stimulatory effect of irradiation dropped precipitously shortly after transfection, but was still observed as late as 24 hours later when circular plasmid DNA was used (S1E and S1F Fig). We also performed an experiment in which the order of transfection and irradiation was reversed, and still observed a stimulatory effect (Fig 1G). This demonstrates that S-RI is caused by the effect of IR on the host cell rather than on the transfected DNA, further supporting the notion that the stimulation is achieved by DSB induction in the genomic DNA. By modifying the amount of DNA electroporated into mES cells we revealed a striking and opposing effects on S-RI and background RI (Fig 1H, S1G and S1H Fig). When DNA amounts were low (2 μg), S-RI was very efficient, and background RI was low, while when high amount of DNA was electroporated (100 μg), S-RI became inefficient, and background RI increased.

## S-RI requires γH2AX but not HR or NHEJ

A simple explanation to S-RI is that radiation creates DSBs into which transfected DNA can be ligated. This model predicts that cells in which major DSB repair mechanisms, such as homologous recombination (HR) and non-homologous end joining (NHEJ), are disrupted, will be more responsive to S-RI, unless the repair deficiency negatively affects the enzymes involved in integration itself. We performed the S-RI assay in mES cell lines where key DNA damage repair and response genes were genetically inactivated. Surprisingly, we observed wild-type dose response curves in $Rad54^{-/-}$ ES cells, which have an HR deficiency phenotype [43], and in $p53^{-/-}$ cells lacking key DNA damage signaling mediator, while in $DNA\text{-}PKcs^{-/-}$ ($Prkdc^{-/-}$) cells deficient in a canonical NHEJ enzyme [44] a small reduction in S-RI was observed (Fig 2A). We recently demonstrated that in mES cells DNA polymerase θ is responsible for the majority of background RI [45], and canonical NHEJ proteins Ku70, Ku80 and LigIV become important in the absence of Pol θ; and similar observations were reported in human cells [46]. Low dose irradiation did not stimulate RI in double-deficient ($Polq^{-/-}$Ku70/80$^{-/-}$) cells (S1 Table), but surprisingly, S-RI was increased rather than suppressed in the Pol θ-deficient cells ($Polq^{-/-}$), and in the Ku mutants (Fig 2B), which suggests that S-RI and background RI have distinct genetic dependencies.

In sharp contrast, cells deficient for H2AX [47,48], a histone variant whose post-translational modifications are central to DNA damage response signaling, were near-completely immune to RI stimulation by irradiation (Fig 2A). This was difficult to reconcile with the previously described phenotypes of $H2ax^{-/-}$ cells, which are prone to translocations [49,50] suggesting increased frequency of DSBs in the absence of a major NHEJ defect. Background RI was normal in $H2ax^{-/-}$ lines (S2A Fig), further indicating that RI and S-RI are genetically distinct processes.

The S-RI defect was observed in two different $H2ax^{-/-}$ cell lines and could be reverted by inserting a wild-type copy of $H2ax$ into the $Rosa26$ "safe harbor" locus (Fig 2C, S2B and S2C Fig). We also tested variants of H2AX with mutations in residues that are phosphorylated (S139) or ubiquitinated (K13, K15, K118, K119) during DNA damage response signaling [51–53], as well as Y142 required for interaction with the key downstream effector MDC1 [54]. Mutations in the lysines– these residues are common to all H2A variants– did not impair S-RI, while S139A and Y142A mutants were indistinguishable from uncomplemented $H2ax^{-/-}$ cells (Fig 2C).

To determine if S-RI was permanently blocked or just stunted by $H2ax$ inactivation we performed a broad-dose S-RI experiment (Fig 2D, S2D Fig), using doses up to 3 Gy. The gradual increase in S-RI efficiency we observed in $H2ax^{-/-}$ cells in this dose range indicated that the end joining reaction responsible for the ligation of extrachromosomal DNA into the ionizing radiation-induced DSB was inherently functional, but less efficient in the absence of H2AX. $H2ax^{-/-}$ cells were also deficient for RI stimulation by etoposide (Fig 2E), one of DNA topoisomerase II inhibitors, which were previously shown to be potent RI stimulators [24,55,56].

Finally, we wondered, which of the checkpoint kinases [57] phosphorylating H2AX at the sites of DNA damage is involved in S-RI. Therefore, we performed experiments in the presence of chemical inhibitors: KU-55933 specific for ATM [58], VE-821 specific for ATR [59], wortmannin that primarily affects DNA-PKcs [60], and the Chk1 inhibitor UCN-01 [61] at the concentrations we previously found to be effective in mouse ES cells [62]. We also tested the effect of caffeine, which is widely used as a broadly specific ATM/ATR/DNA-PKcs inhibitor, but which we found to lack this activity in ES cells, and to strongly suppress gene targeting by HR [62–64]. Inhibition of ATM and ATR reduced S-RI efficiency (S2E and S2F Fig), and the combination of the two inhibitors had an additive effect, which was even more pronounced in

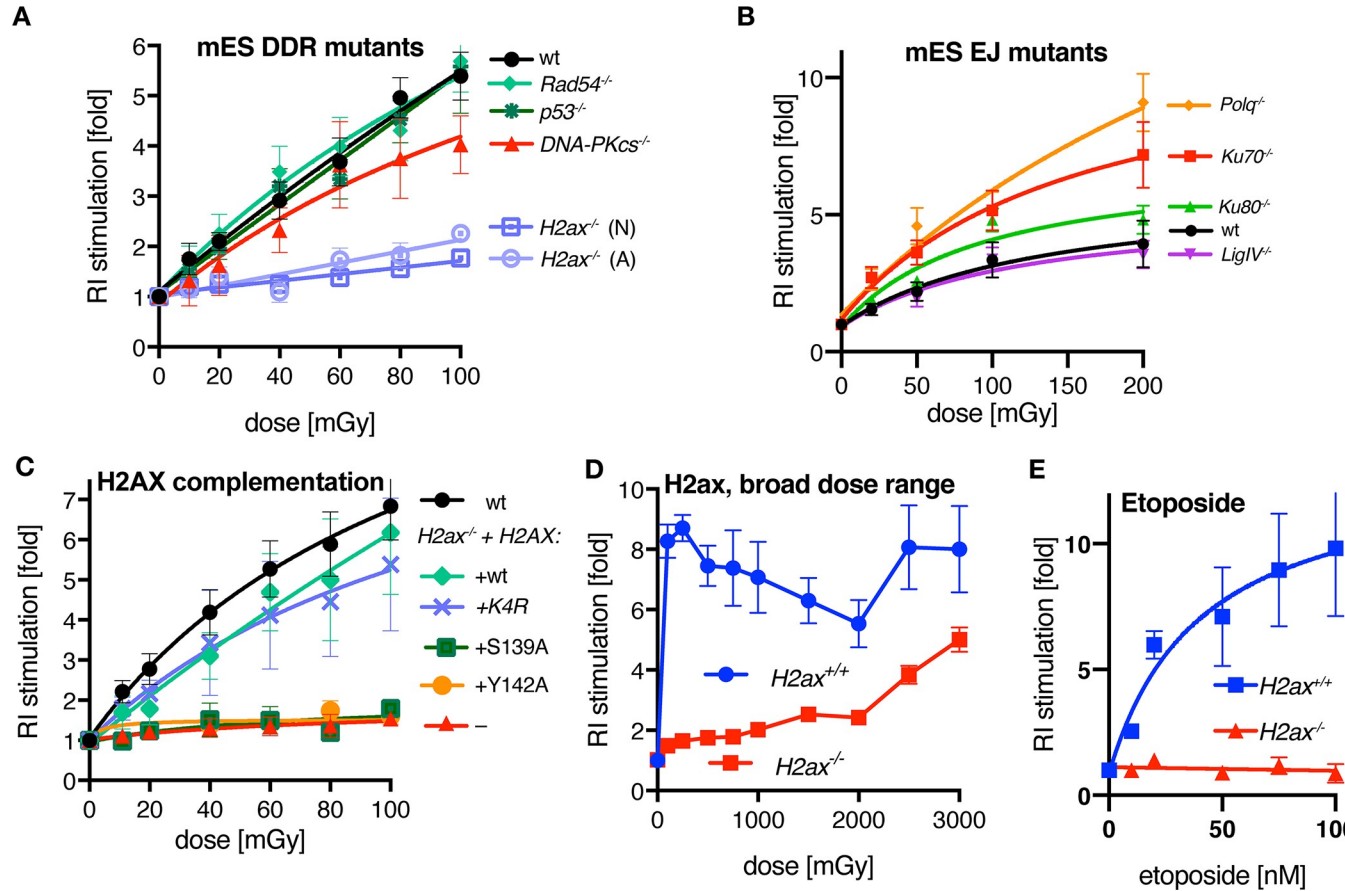

**Fig 2. Genetic dependencies of the IR-stimulated random integration.** (**A**) Colony-based S-RI assay performed with mES cells deficient for key DSB repair and DNA damage response (DDR) proteins. The numbers of colonies obtained after puromycin selection were normalized to the unirradiated control. Background RI frequency is reported in S2A Fig. Means from at least 3 independent experiments, fitted with sigmoid curve, are plotted; error bars show s.e.m. (**B**) mES lines deficient for end joining proteins were assayed as in panel (A). Background RI frequencies for these lines were reported previously [45]. (**C**) *H2ax^-/-* (N) cells were complemented with versions of H2AX containing mutations in the residues involved in key post-translational modifications during the DNA damage response. A single copy of the respective *H2AX* genes was inserted in to the *Rosa26* locus. n = 3, s.e.m. K4R designates a mutant in which four lysines subject to ubiquitination were replaced with arginines. Background RI frequencies for this experiment is reported in (S2C Fig). (**D**) S-RI response to broad irradiation dose range of *H2ax^-/-* and wild-type mES cells was measured in a colony-based S-RI assay. Each point represents means of at least six biological replicas, adjusted for reduced survival (S2D Fig), error bars indicated s.e.m. Both (N) and (A) *H2ax^-/-* lines were used. (**E**) RI stimulation by etoposide measured by colony formation S-RI assay. Experiment was performed twice, with two independent *H2ax^-/-* lines in each ((A) and (N)), data for each genotype was averaged, error bars indicate s.e.m.

*DnaPKcs^-/-* cells (S2E and S2F Fig). In contrast, treatment with UCN-01, caffeine or wortmannin had no effect on S-RI. Taken together these results indicate that S-RI is dependent on phosphorylation of H2AX by one of the partially redundant DNA damage response kinases.

## Role of γH2AX-binding proteins

Three proteins have been shown to directly bind H2AX in phospho-S139 dependent manner: MDC1 [54], 53BP1 [65] and MCPH1 [66]. We engineered a series of mES knock-out cell lines (Fig 3, S3–S5 Figs) to test if S-RI deficiency in cells that cannot form γH2AX is due to the failed recruitment of these proteins. Based on its role in promoting DSB repair via NHEJ over HR, 53BP1 would be a good candidate for the role of the downstream effector, however neither the knock-out nor shRNA knock-down of 53BP1 affected S-RI or background RI (Fig 3A, S3A, S3B and S4 Figs). The most pronounced phenotype of MDC1-deficient cells was a significant

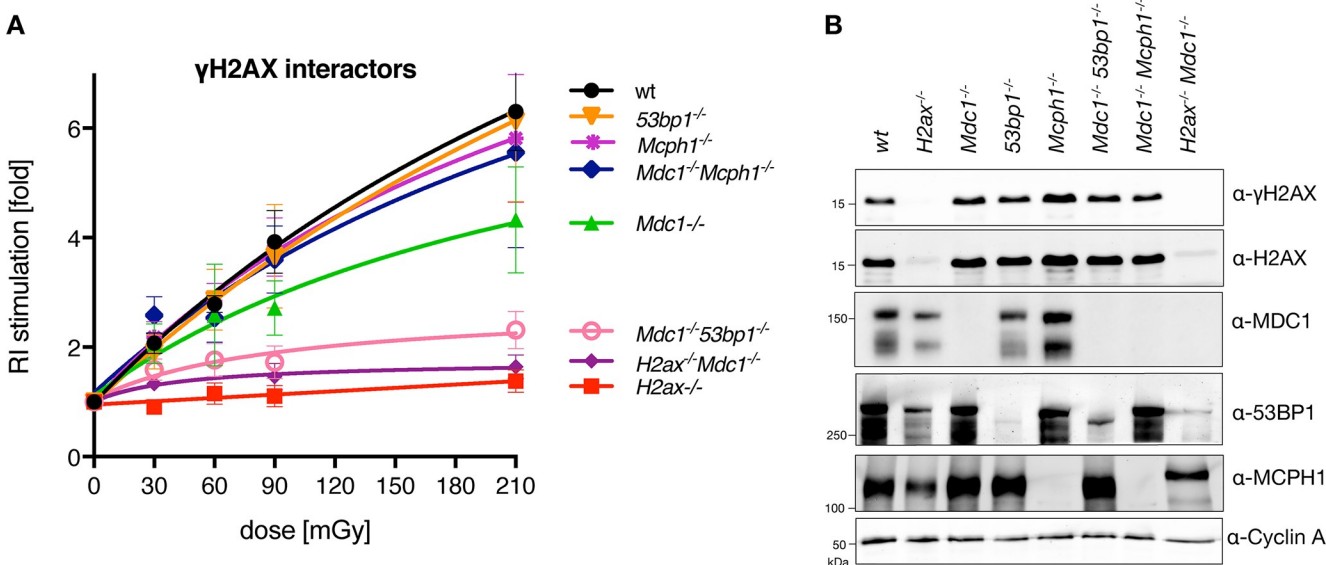

**Fig 3. S-RI dependence on γH2AX-interacting proteins.** (**A**) Colony formation-based S-RI assay was performed on mES cells in which genes encoding the three known γH2AX-interacting proteins were inactivated by CRISPR/Cas9-assisted gene targeting. Background RI frequencies for this experiment are reported in (S3B Fig). (**B**) Immunoblot on the whole cell lysates from the knock-out lines used in panel (A), confirming the loss of protein expression from targeted genes.

reduction in background RI, similar in magnitude to what we reported previously for Pol θ [45]. However, unlike in Pol θ-deficient cells (Fig 2B), S-RI was partially suppressed in $Mdc1^{-/-}$ cells (Fig 3A), and reduced even further in the double $Mdc1^{-/-}53bp1^{-/-}$ knock-out line. Inactivation of $Mdc1$ in $H2ax^{-/-}$ cells did not further suppress S-RI, consistent with the downstream role of MDC1 in γH2AX signaling. $Mcph1$ knock-out cells behaved like wild-type in the S-RI assay, but had elevated background RI. The $Mcph1$ mutation also reverted the partial suppression of S-RI by $Mdc1$ inactivation, as $Mdc1^{-/-}Mcph1^{-/-}$ double mutant showed the same S-RI efficiency as wild-type cells. Since we established that MDC1 contributes to both RI and S-RI efficiency, the positive effect of MCPH1 deletion can be explained by its competition with MDC1 for phospho-S139 binding.

## Discussion

We used a simple assay to re-examine a phenomenon known since the early days of research on cultured eukaryotic cells [22,23]. This makes a number of unexpected observations regarding radiation-stimulated integration of extrachromosomal DNA we present here all the more interesting. The most important of them are: high sensitivity to very low, physiologically relevant doses of irradiation, non-linear dose dependence, independence of the best known DSB repair pathways, NHEJ and HR, requirement of γH2AX for stimulated but not background RI, and the complex involvement of γH2AX-interacting proteins.

Despite numerous reports describing the stimulation of RI by ionizing radiation, to our knowledge our study for the first time reports the effects of doses <0.5 Gy, and in particular the striking dose-dependent stimulation in the range 0.01–0.2 Gy, suggesting that this is a highly efficient process (S1 Text). Previous studies performed with a dose range of 1–10 Gy with different vertebrate cell lines, vectors and transfection methods, observed linear dose dependence of stable colony numbers after adjustment for ionizing radiation-induced reduction in survival [12,15,18,19,22,23]. Explanation of the dose-response curve plateauing we

observed in the majority of experimental systems we used (Fig 1, S1 Fig), and seen in some previous studies [14,17], requires a model that considers more than just chromosomal DSBs as the factor limiting extrachromosomal DNA integration. What can be the other bottleneck(s)? Transfection efficiency is clearly not one, as the frequency of stable integration events ($10^{-2}$–$10^{-4}$) is orders of magnitude lower than the number of transfected cells ($>10^{-1}$). Furthermore, we observed that decreasing the amount of transfected DNA makes S-RI *more* efficient. The nature of the barrier limiting S-RI, which may be cell-intrinsic (e.g. capacity of a DNA-binding protein, signals triggered by DSBs or transfected DNA above a certain threshold), or reflect heterogeneity in the cell population, remains to be determined.

### γH2AX as a central factor involved in stimulated random integration

We observed a striking and unanticipated difference in S-RI between the wild-type and the *H2ax*$^{-/-}$ ES cells. Although γH2AX is a widely used DNA damage marker and actively studied as such [reviewed in ref. 67], the effects of H2AX deficiency on cells and organisms are generally described as "moderate" or "mild" [47,48,68–71], and (S2D Fig). The near-absolute dependence of S-RI on γH2AX is among the most striking H2AX phenotypes discovered to date. Several ways a DNA repair protein may affect RI can be envisaged: by direct involvement in the NHEJ reaction, indirectly by influencing DSB repair pathway choice or facilitating NHEJ protein recruitment etc., or by changing DSB persistence (longer half-lives due to inefficient repair will increase the probability of encounter between a genomic DSB and an extrachromosomal DNA molecule). Other indirect effects through chromatin mobility or packing (accessibility) can also be considered.

H2AX contributes to efficient repair of DSBs by NHEJ. H2AX-deficient cells have increased frequency of both background and induced chromosomal aberrations [47,48,70]. V(D)J recombination although superficially normal in *H2ax*$^{-/-}$ mice [47,48] has hidden alterations revealed by additional inactivation of p53, Artemis or XLF [67]. On the other hand, sensitivity of H2AX-deficient cells to ionizing radiation ranges from moderate [47] to marginally detectable [70], and is always lower than core NHEJ mutants [47]. Moreover, some of this increased sensitivity can be attributed to impaired HR [48,70,72,73] and a compromised G2/M DNA damage checkpoint [70,74]. Assays monitoring mutagenic repair of nuclease-induced DSBs in a chromosomal reporter revealed no effect of H2AX deletion [75], but recent studies described a role of H2AX in mutagenic NHEJ of Cas9-induced DSBs [76,77]. Data on DSB clearance kinetics in H2AX-deficient cells is conflicting (refs [48,68] vs [78]). Thus, previously described involvement of H2AX in NHEJ does not provide a compelling explanation to S-RI suppression.

Among γH2AX-binding proteins 53BP1 was a prime candidate for the role of the γH2AX-dependent S-RI mediator [79,80], as its retention at DSBs is γH2AX-dependent [71], it promotes DSB repair by NHEJ over HR [81], facilitates of the synapsis between distal DSBs [82] and increases the mobility of a chromosomal DSBs [83]. However, we could only observe the effect of 53BP1 deletion on S-RI when MDC1 was also absent. Moreover, we found that MDC1 rather than 53BP1 contributes to background RI, which was also observed previously in human cells [54,84], and deletion of 53BP1 had no significant effect even when combined with *Mdc1* inactivation. While several studies showed that 53BP1 recruitment to ionizing radiation-induced foci is controlled by MDC1 [85–87], our results suggest an MDC1-indepenent role of 53BP1, as double mutation *Mdc1*$^{-/-}$*53bp1*$^{-/-}$ is required to recapitulate the effect of H2AX deficiency on S-RI.

### Are S-RI and RI distinct processes?

The S-RI and background RI have distinct genetic dependencies: while γH2AX is required for S-RI, its loss does not affect background RI; Pol θ has an opposite effect: it is responsible for

the majority of background RI events, but its deletion does not impair S-RI; inactivation of *Mdc1* severely impairs background RI [54,84], but has a much smaller effect on S-RI; *Mcph1* knock-out increases RI efficiency, but has no effect on S-RI unless *Mdc1* is also inactivated. These observations can be interpreted as an indication that S-RI and RI are mechanistically different. However, the models based on this supposition and accommodating the observation we present here and previously [45] are inevitably complex (S3C Fig), and require some uncomfortable assumptions. For example, we found that two out of three γH2AX-binding proteins we studied affect background RI (MDC1 stimulates, MCPH1 suppresses (S3B Fig)). If we postulate that background RI itself is γH2AX-independent, we need to conclude that each of the two proteins is coincidentally recruited in some γH2AX-independent manner.

The alternative is to suppose that at least some of the apparent genetic distinctions between RI and S-RI are misleading, or that they stem from the factors that are beyond the simple "[DSB] • [ecDNA] → end joining → insertion" model. As an example of the former, the concentration of background DSBs in the $H2ax^{-/-}$ cells (known to be genetically unstable), could be so high as to reach the saturation state that is responsible for the plateauing of the dose-response curve. However, our broad-dose experiment argues against this explanation (Fig 2D). Factors outside of the DSB repair paradigm may include alteration of the cellular state due to DNA damage and antiviral defense checkpoint inductions we alluded to in the discussion of the dose-response plateau, role of H2AX and its interactors in transport, chromatinization, persistence and integrity of the ecDNA, etc. Further genetic exploration of the RI and S-RI phenomena should provide important clues. It is particularly interesting to trace the connection between the upstream signaling proteins (H2AX, MDC1) and the end-joining proteins responsible for the insertion reaction (Pol θ and canonical NHEJ).

### Implications

Our results have several important implications. The S-RI assay is macroscopic and thus simpler than direct and surrogate mutagenesis assays and damage detection methods (DNA damage response foci counting), and is as sensitive as the alternatives. A high throughput version of it can be developed for determining chemical genotoxicity at physiologically relevant concentrations *in vitro*. Unlike the surrogate methods, the assay detects irreversible mutagenic events, rather than damage that may or may not be repaired. Analysis of the genetic dependencies of S-RI led to several unexpected findings; this is arguably the most pronounced phenotypic manifestation of H2AX deficiency, and thus the assay is a useful tool to study functional interactions between the components of the convoluted γH2AX signaling pathway, NHEJ and other DSB repair proteins.

Although we used several different human and mouse cell lines, forms of extrachromosomal DNA and radiation sources, it is important to stress that further experiments using organoid and animal models, and perhaps epidemiological studies, will be required to determine if our findings have physiological or public health relevance. If the phenomenon is not limited to cultured cells, episomal DNA concentration–for example from persistent infection with viruses such as HPV or HBV–may need to be considered as a confounding factor in radiation risk assessment.

## Materials and methods

### Cell lines

IB10 mouse ES cells, a clonal isolate of the E14 line [88], and other mouse ES cell lines used in the study were maintained on gelatinized plastic dishes as described [62]. ES cells were grown in 1:1 mixture of DMEM (Lonza BioWhittaker Cat. BE12-604F/U1, with Ultraglutamine 1, 4.5

g/l Glucose) and BRL-conditioned DMEM, supplemented with 1000 U/ml leukemia inhibitory factor, 10% FCS, 1x NEAA, 200 U/ml penicillin, 200 μg/ml streptomycin, 89 μM β-mercaptoethanol. $H2ax^{-/-}$ (A) and (N) ES cells were kindly provided by the Alt and Nussenzweig laboratories, respectively [47,48]. HeLa and U2OS cells were grown in DMEM, 10% FCS, 200 U/ml penicillin, 200 μg/ml streptomycin. AID-AsiSI-ER U2OS cells were a king gift of Gaëlle Legube [42]. HEK293T cells were grown in DMEM, 5% FCS, 200 U/ml penicillin, 200 μg/ml streptomycin.

## Generation of knock-out ES cell lines with CRISPR/Cas9

*Mdc1*, *Mcph1*,*53bp1* knock-out ES cells were produced by CRISPR/Cas9 stimulated gene targeting with plasmid donor (S4 and S5 Figs). The CRISPR/Cas9 expression plasmid (derived from pX459 [89]) contained one or two sgRNA expression cassettes. Target sequences were (PAM underlined): for *Mdc1* #2 AAGGTAGAGGGGGAAATCTG<u>AGG</u> and #3 AACAGTA GTTCCAGAAAGGT<u>GGG</u> within exons 3 and 4; for *53bp1* #1 TAGTTGAGGTCGGCTTGA GG<u>TGG</u> upstream of the promoter and #2 CCATCAGTCAGGTCATTGAAC<u>GG</u> within exon 4; for *Mcph1* promoter targeting #1 CCGGCGCTTAAGGCGACGAAA<u>GG</u> and #2 AAAGCA ACTTGAGGATATGG<u>GGG</u>, for *Mcph1* exon 4–5 #3 TGTTCATCGGTATTCACTGC<u>AGG</u> and #4 TGTGCCTGACAGCTACAGGG<u>AGG</u>. Donor constructs contained PGK-hyrgo (*Mdc1*) or PGK-neo (*Mcph1*, *53bp1*) selection cassettes. $Mdc1^{-/-}53bp1^{-/-}$ and $Mdc1^{-/-}Mcph1^{-/-}$ cells were produced from the $Mdc1^{-/-}$ cell line. For each genotype two to four independent clones were used in experiments.

For *H2ax* complementation NotI-linearized *Rosa26* [90] targeting vectors containing the human H2AX CDS under the *H2ax* promoter and with *H2ax* 3'UTR were electroporated into ~1–2×10$^7$ $H2ax^{-/-}$(N) ES cells. The vectors were derived from *Rosa26* gene targeting vector pHA416 (a kind gift from Hein te Riele) containing *Salmonella typhimurium* hisD coding sequence. Two days after electroporation selection with 2 or 4 mM L-hisitidinol (Sigma, H6647) was started. Media was replaced every 2–4 days for 10–14 days; colonies were picked and expanded. Single copy integration into the *Rosa26* locus was confirmed by DNA hybridization on BamHI-digested genomic DNA with a 5' probe (PstI-SalI fragment, from pHA607 [91]). H2AX expression was verified by immunoblotting with anti-γH2AX (Millipore mouse mAb clone JBW301) or anti-total H2AX antibody (Cell Signaling rabbit polyclonal #2595).

## Constructs

Constructs used in this study were generated by homology-based cloning methods: SLIC [92], In-Fusion (Clontech), isothermal Gibson assembly [93] or recombineering with mobile reagents [94]. Pfx50 polymerase (Invitrogen) was used for PCR amplification of the construction elements in most cases, Phusion (Finnzymes) or Platinum Taq (Invitrogen) polymerase were used to amplify from genomic DNA or cDNA. Constructs were partially sequenced to verify the ligation junctions and ensure the absence of PCR-induced mutations. Maps and details are available upon request.

pLPL, the construct used in the majority of S-RI experiments, was derived from the construct loxP-PGK-gb2-neo-polyA-loxP cassette in pGEM-T Easy originally designed to engineer knock-out alleles using recombineering (Francis Stewart lab, distributed at 2007 EuTRACC workshop on recombineering), referred to as pLNL. A dual (bacterial (gb2) and eukaryotic (PGK)) promoter drives the antibiotic resistance gene allowing selection in both hosts. Neomycin phosphotransferase (*neo*) was replaced with puromycin N-acetyltransferase (*pac*, puro) or hygromycin phosphotransferase (hygro) using recombineering to produce

pLPL and pLHL, respectively. For linearization pLPL was digested with DraI, phenol-extracted, precipitated with isopropanol and dissolved in deionized water.

*H2ax* complementation constructs used to insert a single copy of the human or mouse *H2AX* CDS under native mouse *H2ax* promoter and 3'UTR were engineered by replacing the backbone and removing the PGK-polyA cassette from the pHA416 vector (a gift from Hein te Riele) to create a unique XhoI site between the histidinol trap cassette and the 3' homology arm (pAZ025). A 1721-bp fragment of the mouse *H2ax* locus was PCR-amplified using H2AXwhole-F/-R primers first into a shuttle vector, and then re-cloned into pAZ025, resulting in pAZ026. Analogous constructs were engineered with human H2AX and its mutants by PCR amplification of the promoter and UTR regions from pAZ026 and coding sequences from the corresponding pMSCVpuro constructs [53] (pAZ080-pAZ085); Y142A mutation was added by site-directed mutagenesis (pAZ122).

### S-RI assays

Drug resistant colony formation was used as a measure of stable RI frequency in the majority of experiments. In a typical experiment $6 \times 10^6$ mouse ES cells were electroporated in 450 µl growth media with 10 µg circular or linearized pLPL (puro) or pAZ095 (GFP, puro) plasmid DNA using GenePulser Xcell apparatus (118 V, 1200 µF, $\infty$ Ω, exponential decay). pEGFP-N1 plasmid (10 µg) was co-electroporated with pLPL to estimate transfection efficiency in the experiments, where absolute targeting efficiency was determined. In the initial experiments electroporated cells were re-suspended in 37–68 ml growth media; the suspension was distributed over 5–8 10 cm dishes at 7 ml per dish. In later experiments, involving larger number of cell lines, electroporated cells were re-suspended in media and equally distributed into 1.5 ml microcentrifuge or 0.2 ml PCR tubes at 0.1–1 ml per tube. Cells were irradiated within 1 hr after transfection with different doses using $^{137}$Cs source (Gammacell) or X-ray apparatus (RS320, Xstrahl). A metal attenuator reducing the dose rate by ~50% was used to deliver doses <100 mGy in low-dose irradiation experiments with Cs source. For micro-CT irradiation cells in 6 cm culture dishes or in 1.5 ml microcentrifuge tubes were scanned (Quantum FX, Perkin Elmer) in the low resolution mode (73 mm f.o.v., 17 sec, 148 mm pixel size), repeatedly 1–5 times. The initial number of cells used for electroporation was increased to compensate for low plating efficiency of some knock-out cell-lines (e.g. *H2ax*$^{-/-}$ and their derivatives). After irradiation in microtubes cells were plated in 10 ml media in 10 cm dishes. To estimate plating efficiency at various irradiation doses 2–4 µl aliquots of irradiated cell suspension was plated into 6-well plates in triplicates. Puromycin (1.5 µg/ml) selection was started one day after electroporation; media was changed as required until macroscopically visible colonies were formed. Colonies were washed with PBS, stained with Coomassie Brilliant Blue (100 mg/l in 40% methanol, 10% acetic acid), and counted directly or after photographing.

Background RI frequency was determined by dividing colony counts in unirradiated control plates by the number of viable cells plated, and adjusted for transfection efficiency. RI stimulation was determined by dividing the number of colonies in irradiated plate by the number of colonies in unirradiated control. Because the low doses of irradiation used in most experiments do not induce significant cell death, correction for reduced viability due to irradiation was not necessary and was only performed for broad dose irradiation experiments and with acutely IR-sensitive mutants. Background RI shown in supplementary and S-RI plots shown in the main figures represent all data collected in the low dose irradiation experiments. To produce the low dose S-RI plots, normalized data from repeat experiments was averaged and fitted with a sigmoid curve using the "[Agonist] vs. response" function in GraphPad Prism software v8. Statistically significant differences in background integration frequency between

wild type and mutant lines were determined by one-way ANOVA with Dunnett's multiple comparisons test and are indicated with asterisks (* $p \leq 0.05$, ** $p \leq 0.01$, *** $p \leq 0.001$, **** $p \leq 0.0001$).

Experiments where cells were irradiated before electroporation were performed either with attached cells growing in 6-well plates or in suspension. For irradiation in suspension cells growing in 145 mm dishes were collected by trypsinization and re-suspended in growth media at 12.5 million per ml. The suspension was divided in two parts one of which was irradiated with 0 or 200 mGy and transferred to a 10 cm culture dish, which was kept in a cell culture incubator during the time course. At various time points after irradiation cell suspension was mixed by pipetting to prevent attachment, a 400 μl aliquot was taken and electroporated with 5 μg linear pLPL DNA. For irradiation in plates cells were seeded at 4 million per well into 6-well plates (two per each time point) one day before the experiment. Next day half of the plates were irradiated with 200 mGy. At various time points after irradiation cells from one well from irradiated and unirradiated plate was trypsinized, collected by centrifugation, resuspended in 400 μl growth media and electroporated with 5 μg linear pLPL DNA. Electroporated cells were plated in 10 cm dishes; an aliquot was transferred to a 6-well plate for plating efficiency estimation; puromycin selection was started in 10 cm dishes one day after seeding.

S-RI experiments with *Polq* and *Ku70/80* double deficient cells were performed by combining multiple electroporation reactions (10 million, 10 μg linear pLPL per electroporation, 13–24 electroporations per experiment) into a single cell suspension, dividing it in two unequal parts, one of which (2/3) was irradiated with 100 mGy. Cells were then seeded into gelatinized 145 mm culture dishes; selection was started next day. Aliquots were taken after seeding from each dish and grown in 6-well plates without selection to determine plating efficiency; a plate of untransfected cells (number equal to the number of cells in unirradiated sample) was used as a selection efficiency control. Plates were stained 8–10 days after transfection.

To determine the effect of DNA amount, 400 μl of $15 \times 10^6$ per ml cell suspension were electroporated with 2, 5, 10, 50 or 100 μg linearized pLPL DNA and diluted in different volumes (1, 1, 2.5, 5, 8 ml, respectively) of media to account for differences in background RI and produce approximately similar colony densities in unirradiated dishes; 150 μl aliquots of the diluted suspension were distributed into 8 PCR tubes, which were irradiated with 0–500 mGy and seeded into 10 cm dishes with 10 ml growth media; 5 μl and 50 μl aliquots were taken from the dishes irradiated with lowest and highest doses, seeded in triplicates into 6-well plates and used to estimate plating efficiency without selection and clonogenic survival. Colony numbers in 10 cm dishes after selection were adjusted for dilution and survival, and normalized to unirradiated control as in other experiments. Two independent experiments were performed.

For etoposide treatment, electroporated cells were seeded into 10 cm dishes containing a range of etoposide (Sigma E1383, stock solution 10 mM in DMSO) concentrations in 10 ml media; after seeding 5 μl and 50 μl aliquots were taken and plated in triplicate into 6-well plates containing 2 ml media with the same etoposide concentration; media was replaced next day and selection started in 10 cm dishes. Colony counts were adjusted for survival and normalized to untreated control. In the experiments involving kinase inhibitors, the stock solutions of chemicals (Ku-55933 10 mM in DMSO, VE-821 10 mM in DMSO, caffeine 40 mM in ES media or 100 mM in water, UCN-01 100 μM in DMSO, Wortmannin 1 mM in DMSO) were added to the media after seeding; 6 hr later media was collected and replaced with fresh media; collected media was centrifuged to pellet the cells, which were returned to the dish.

For FACS-based S-RI assays cells were electroporated with pEGFP-N1 or pAZ095, plated into 6 cm dishes at $3–6 \times 10^5$ per dish and irradiated. One or two days later the percentage of GFP-positive cells was determined (transient transfection efficiency). Upon reaching confluence cells were replated at 1:5–1:10 dilution into 6-well plates; this was repeated until 10–14

days after transfection at which point the percentage of GFP-positive cells (RI frequency) was measured.

## Immunoblotting

Immunoblotting was performed following standard protocol by wet transfer to nitrocellulose or PVDF membrane, blocking and antibody dilution solution contained 5% dry skim milk, 0.05% Tween 20 in PBS. Secondary antibodies were either HRP-conjugated detected with ECL (GE), or fluorescent (Sigma, anti-mouse CF680 SAB4600199, anti-rabbit CF770 SAB4600215) detected with Odyssey CLx scanner (LiCOR). Primary antibodies: 53BP1 (Novus Biologicals, NB100-304), H2AX (Cell Signaling, #2595), γH2AX (pS139, Millipore JBW301), MDC1 ("exon8", Abcam, ab11171 and P2B11, Millipore, 05–1572), MCPH1 (Cell Signaling, #4120), Cyclin A (Santa Cruz, C-19), PARP-1 (ENZO, C-2-10), Chk2 (BD Transduction, #611570).

## Virus production

For rAAV production a confluent 10 cm dish of HEK293T cells was trypsinized and seeded at 1:2 dilution into a fresh 10 cm dish with 17 ml growth media. Calcium phosphate transfection was performed by mixing 10 μg each of the packaging (pHelper and pAAV-RC2) and 10 μg rAAV genome encoding GFP (pAAV-GFP) plasmids, 100 μl 2.5 M $CaCl_2$, deionised water to 1 ml; then 1 ml 2xHBS (16.4 g/l NaCl, 11.9 g/l HEPES, 0.21 g/l Na2HPO4; pH7.1 with NaOH) was added while bubbling air through the solution. The transfection mix was added dropwise to the cells. Media was changed next day and $10^6$ HeLa cells were seeded for infection. Two days after transfection HEK293T cells were washed and dispersed with PBS containing 10 mM EDTA, pelleted, re-suspended in 1 ml media, frozen on dry ice/ethanol bath and thawed at 37˚C; the freeze-thaw cycle was performed the total of four times. Lysate was centrifuged at 10,000 rcf for 10 min. Half of the lysate was used to infect the HeLa cells. One day after infection HeLa cells were trypsinized and counted, divided over 10–15 10-cm dishes, which were irradiated with various doses. An aliquot was analyzed by FACS to determine the transduction efficiency. On days 5–8 cells were checked and passaged if confluent. FACS analysis to determine the fraction of cells still expressing GFP was performed on days 8 to 13.

D64V mutation inactivating the integrase was introduced into lentiviral packaging plasmid pMDLg/pRRE by replacing the AgeI-AflII fragment with two overlapping PCR products, with mutation in the overlap, using Gibson assembly (pAZ139). HEK293T cells were transfected by calcium phosphate precipitation as described above with the third generation lentiviral packaging constructs pRSV-Rev, pMDLg/pRRE D64V (pAZ139) [95], and the plasmid encoding lentiviral genome with PGK-puroR (pLKO.1) alone or additionally with CMV-TurboGFP (SHC003, Sigma). Virus containing media was collected on days 2 and 3 after transfection, diluted 1:2 with the appropriate growth media and used to infect IB10 or HeLa cells (2x 10 cm-dishes). One day after the second infection the target cells were collected, pooled, distributed over three 145-mm dishes and irradiated with 0, 100, 400 mGy. Puromycin selection was started one day after irradiation.

## Supporting information

**S1 Fig. Supporting data for Fig 1.** (**A**) S-RI in HeLa cells transfected by electroporation with linear or circular plasmid DNA carrying puromycin resistance gene under a PGK promoter. (**B**) S-RI in HeLa cells transfected by lipofection. (**C**) Representative FACS plots from the rAAV S-RI experiments shown in (Fig 1D). HeLa cells infected with rAAV2-GFP virus and analyzed at different time points after infection. Gates P4 and P7 were used to calculate the fraction of high GFP-positive and all GFP-positive cells, respectively. The latter better reflects

transient transduction efficiency, while the former more reliably represents the population of cells that still continuously produce the GFP protein. Percentage of GFP-positive cells drops precipitously from day 2 to day 8. At 8 days post infection the broader gate (P7) still contains cells that express GFP from transient infection, while the more stringent gate contains cells stably expressing GFP. (**D**) FACS-based S-RI assay performed with U2OS cells electroporated with the GFP-encoding plasmid DNA. (**E**) Stimulation of RI in HeLa cells electroporated with circular or linear plasmid DNA and irradiated with 0.2 Gy at different time points after electroporation. (**F**) Stimulation of RI in mES cells electroporated with circular or linear plasmid DNA and irradiated with 1 Gy at different time points after electroporation. Colony numbers were adjusted for reduced viability due to irradiation, and normalized to unirradiated control. (**G**) Repeat of the experiment shown in (Fig 1H). (**H**) Background RI frequency (number of puromycin-resistant colonies per viable cell plated) from the experiments reported in (Fig 1H, S1G Fig). (PDF)

**S2 Fig. Genetic dependencies of S-RI (related to Fig 2).** (**A**) Background RI in the mutant cell lines used in S-RI assay from (Fig 2A). Individual values from biological replicas are plotted, with bars indicating means ±s.e.m. Statistical significance was determined using one-way ANOVA with Dunnett's multiple comparison test. (**B**) Immunoblot of $H2ax^{-/-}$ and complemented lines. Total cell lysates from wild-type, $H2ax^{-/-}$ (A) and (N) lines, and $H2ax^{-/-}$ (N) line complemented with various H2AX mutants, were immunoblotted with the indicated antibodies. To test γH2AX induction cells were irradiated with 4 Gy and lysed 30 minutes after. (**C**) Background RI measured as in panel (A) in cells used in (Fig 2C). (**D**) Samples from plates used in the S-RI assay plotted in (Fig 2D) were taken to determine the effect of irradiation on clonogenic survival of the wild-type and $H2ax^{-/-}$ cells to compensate for loss of viability in S-RI. (**E**) Effect of DNA damage response kinase inhibitors on S-RI in wild-type (black circles) and $DNA-PKcs^{-/-}$ (red squares) mES cells. Cells were electroporated with linearized plasmid, seeded into dishes containing the indicated concentrations of the inhibitors and irradiated with 50 mGy. The chemicals were removed 6 hours later. Data from four independent experiments is plotted. (**F**) Cells were treated with the compounds used in the experiment shown in panel (E), irradiated with 0 to 10 Gy, lysed 4 h later, and analyzed by immunoblotting with the indicated antibodies. Phosphorylated and unphosphorylated forms of Chk2 are indicated with arrows on the upper blot. (PDF)

**S3 Fig. Supporting data for Fig 3.** (**A**) 53BP1 knock-down does not affect S-RI efficiency. Means of four independent puromycin-resistant colony formation S-RI assays (two with linearized, two with circular plasmid DNA) are plotted. Immunoblot on total cell lysates with the indicated antibodies confirming the efficiency of knock-down is shown as an inset. (**B**) Background RI efficiency in mES cell lines deficient for γH2AX interacting proteins. Data is plotted as in (S2A Fig). (**C**) A model of RI and S-RI, based on the supposition that the initial stages of the two processes are mechanistically distinct ❶, to account for the observation that S-RI is γH2AX-dependent ❷, while RI is not ❸; MCPH1 competes with MDC1 for γH2AX binding, and its removal results in elevated RI ❸. MDC1 contributes to both RI and S-RI ❹, and 53BP1 can provide a backup mechanism for S-RI in the absence of MDC1, but does not contribute to the RI process ❹. Since neither RI nor S-RI are completely abolished by the deletion of the proteins listed in the scheme, alternative pathways must exist ❺. The final ligation steps are mediated by Pol θ or cNHEJ ❻, as we previously showed that all integration events in ES cells are abolished when both these end joining mechanisms are inactivated, however the relative contribution of Pol θ and cNHEJ to RI and S-RI may be different. (PDF)

**S4 Fig. Generation of knock-out mES lines by CRISPR/Cas9-assisted gene targeting.**
Schemes of mouse loci (**A**) *Mdc1*, (**B**) *53bp1* and (**C**) *Mcph1* and of the corresponding gene targeting construct are shown. CRISPR/Cas9 cut sites (gRNA recognition sequences) are indicated with red arrows. Homology arms and PCR primers used to amplify them from genomic DNA for cloning into the gene targeting construct are shown as violet bars and arrows, respectively. Locations of the PCR primers used to screen for homozygously targeted clones are shown as blue arrows. Endogenous and synthetic promotors are shown as thick arrows. Exons and antibiotic resistance genes are shown as bars.
(PDF)

**S5 Fig. Immunoblots confirming the loss of protein expression from targeted genes.** Total cell extracts from the mES cell lines with indicated genotypes were fractionated by SDS-PAGE and immunoblotted with the indicated antibodies: (**A**) anti-MDC1 monoclonal and (**B**) polyclonal raised against the region encoded by exon 8, (**C**) anti-53BP1 and (**D**) anti-MCPH1. Membranes were re-probed with anti-PARP-1 antibody to assess relative loading. For each genotype at least two independent clones used in the experiments were tested. Clones that were derived from the CRISPR/Cas9-assisted gene targeting procedure but retained the wild-type allele (as determined by PCR genotyping) were used as controls in some experiments; these are indicated with letter C. Asterisk indicates non-specific band.
(PDF)

**S1 Table. S-RI in *Polq Ku70/80* double knock-out ES cells.**
(PDF)

**S1 Text. Estimation of the frequency with which DSBs capture transfected DNA.**
(PDF)

## Acknowledgments

We thank Andre Nussenzweig and Frederick W. Alt for *H2ax*^-/- lines, Francis Stewart, Hein te Riele, Titia Sixma and Alan Bradley for providing plasmid constructs, Gaëlle Legube for providing the AID-AsiSI-ER cell line, Yanto Ridwan for help with the micro-CT irradiations.

## Author Contributions

**Conceptualization:** Alex N. Zelensky, Roland Kanaar.

**Funding acquisition:** Dik C. van Gent, Jeroen Essers, Roland Kanaar.

**Investigation:** Alex N. Zelensky, Mascha Schoonakker, Inger Brandsma.

**Resources:** Marcel Tijsterman.

**Supervision:** Alex N. Zelensky, Dik C. van Gent, Jeroen Essers, Roland Kanaar.

**Writing – original draft:** Alex N. Zelensky.

**Writing – review & editing:** Alex N. Zelensky, Marcel Tijsterman, Dik C. van Gent, Jeroen Essers, Roland Kanaar.

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
