## [Decision Letter · Decision Letter 0]

10 Sep 2019

Dear Roland,

Thank you very much for submitting your Research Article entitled 'Low Dose Ionizing Radiation Strongly Stimulates Insertional Mutagenesis in a γH2AX Dependent Manner' to PLOS Genetics. Your manuscript was fully evaluated at the editorial level and by independent peer reviewers. There was a clear consensus that your report an exciting observation with potential implications for medical imaging.

Two of the reviewers also identified scientific issues to be addressed, some minor and some major. The chief question was regarding the effect of a Ku-Polq double mutant, which it seems possible to test in the mES cell line you reported on recently in Nature Communications.

We hope you will decide to revise the manuscript for further consideration here.  The revision should address the specific points made by each reviewer. We will also require a detailed list of your responses to the review comments and a description of the changes you have made in the manuscript.

If you decide to revise the manuscript for further consideration at PLOS Genetics, please aim to resubmit within the next 60 days, unless it will take extra time to address the concerns of the reviewers, in which case we would appreciate an expected resubmission date by email to plosgenetics@plos.org.

[LINK]

We look forward to receiving a revised manuscript. Please do not hesitate to contact us if you have any concerns or questions.

Yours sincerely,

Nancy Maizels, Ph.D.

Associate Editor

PLOS Genetics

Gregory Barsh

Editor-in-Chief

PLOS Genetics

Reviewer's Responses to Questions

**Comments to the Authors:**

Reviewer #1: This is a review of "PGENETICS-D-19-01201." In this manuscript, the authors examine the mechanism of radiation-induced random integration (RI) of DNA, using very low doses of radiation. These findings are highly significant, as the mechanisms of RI have remained poorly understood, and the results could have implications for medical radiation (see below major point 1). The authors use multiple methods to provide convincing evidence that low levels of radiation induce RI using multiple model systems and radiation sources, as well as the AsiSI nuclease. They also examine the mechanisms of radiation-induced RI. For one they find that loss of major DNA repair proteins do not affect RI, although I am surprised that the Ku-Polq- cell line was not tested (major point 2). In contrast, the H2AX-MDC1 signaling factors are shown to be required for radiation-induced RI, including convincing evidence that the signaling residues on H2AX (S139 and Y142) are required for this function. As the precise roles of H2AX and MDC1 in regulating repair remain unclear, it is understandable that further mechanistic insight has been difficult (although recent studies on how these factors influence NHEJ could be included, Minor point 1), but I think that these findings will stimulate research in this area.

Major points.

1. There are potential implications of this study for the ability of medical radiation to cause insertion of extra-chromosomal DNA. This issue is raised a few points in the manuscript, and in the experiment using the micro-CT source. However, as all experiments are performed in cell lines, the implications for patients may be unclear. As this has the potential to have effects on public health, and even possible press attention (which has a tendency to over-simplify concerns), these claims should be carefully stated. For instance, the Discussion does not mention these issues in detail, and the caveats of drawing too many conclusion on implications for medical radiation without clinical data.

2. Given the recent Nature Communications paper from this group with a Polq-Ku- mESC line, I am surprised that this line is not used to examine possible functional redundancy for radiation-induced RI. Perhaps there are technical issues, e.g. because this line is so radiation sensitive, it may be difficult to separate radiation toxicity vs. RI frequency. Nonetheless, I suggest this issue should be addressed.

Minor point.

1. Two recent studies indicate that H2AX and MDC1 may suppress precise canonical non-homologous end joining using Cas9 DSBs, which may be considered in the Discussion when addressing potential mechanisms. PMID: 28057860, PMID: 28977657. In other words, both studies indicate that H2AX is important for mutagenic EJ events, and the latter has evidence with MDC1.

2. Some discussion early in the Results about the dose of linear DNA introduced into cells relative to the mass of chromosomal DNA should be included, even if it is only an estimated range.

Reviewer #2: Zelensky et al investigate an observation that ionizing radiation promotes integration of exogenous DNA into mammalian cell genomes. While this observation has been made previously by other groups, Zelensky show that it is most evident at the low doses, and thus plausibly medically relevant (i.e. after medical imaging). More importantly, this group employs a panel of cell lines defective in members of the DNA damage response to investigate the mechanistic basis for this phenomenon. The work describes some surprising and exciting results, and is for the most part technically well performed. It will be of interest to a wide fraction of this journal’s readers. However, a few critical omissions must be corrected.

Major omissions.

1) The authors determined in previous work that random integration is ablated in cells deficient in both Ku and Polymerase theta. They appear to infer from this prior result that stimulated RI will be similarly dependent on the presence of at least one of the mammalian end joining pathways (Fig S3C). They must formally confirm this, i.e. show that low level radiation is not sufficient to rescue integration in cells deficient in both Ku and Polymerase theta.

2) The authors must confirm efficacy of their checkpoint kinase inhibitor treatments (e.g. using phosphor-specific westerns). This may help resolve apparent contradictory evidence arguing that combined ATRi and ATMi reduced S-RI, even though caffeine treatment, which they suggest inhibits all three kinases, did not. Additionally, the statement “in contrast to UCN-01, caffeine, and wortmannin had no effect on S-RI” appears incorrect. Fig S2E suggests results with UCN-01 are also indistinguishable from untreated.

3) The authors routinely quantify stimulated random integration (“S-RI”) using the fraction “RI stimulation”, as defined by the frequency of random integration after stimulation divided by the frequency of random integration without stimulation. It is often not possible to fully interpret such data. For example, there is an apparent dose-dependent increase in S-RI in cNHEJ mutants; does this simply reflect a decrease in the base line (zero dose) integration frequency in the mutants, or is the relationship more complex? Difficult-to-parse statements of interpretation “Genetic analysis of stimulated RI (S-RI) revealed that it is…not reduced by DNA polymerase θ (Polq) inactivation” vs. “Surprisingly, S-RI was increased rather than suppressed in the Pol θ-deficient cells (Polq-/-), and in the Ku mutants” further add to the confusion. Another figure is necessary for at least one of these mutants, in which the authors plot raw random integration frequencies over the full range in dose (0-100mGy). This would also be helpful in interpreting how increasing transfected DNA amounts impact S-RI, as tested in Figure 1H. Again, it would be helpful to see the relationship between raw integration frequencies for e.g. 2 ug vs. 10 ug transfected DNA over the full range in radiation dose. Simply including the “background” RI frequency (Figure 1I) doesn’t provide all the necessary information.

Minor Concerns

1) The authors leave another significant question unresolved; how long does the irradiation effect last? Regarding Fig 1G, the authors should test a wider range in time.

2) It is difficult to reconcile results in Figure 2B (RI stimulated 2-3-fold at 100mGy in wild type mES), with results in 2A and 2C (RI stimulated 6-7 fold at 100 mGy in wild type mES).

3) The distinction between low (P7 gate) and high (P4 gate) GFP expression in Fig S1C should either be rationalized or omitted.

4) The statement that “wortmannin that has highest specificity for DNA-PKcs” should be more carefully worded, in that it could be interpreted as intending to state that “amongst DNA-PKcs inhibitors, wortmannin has the highest specificity”. There are other DNA-PKcs inhibitors that are more specific for DNA-PKcs.

Reviewer #3: The report by Zelensky and co-workers provides convincing evidence that insertional mutations can occur following radiation doses that are in the range of medical imaging modalities. They also show that the process is more efficient at low extrachromosomal DNA levels and that radiation-induced (or simulated) random insertions may be mechanistically distinct from background (un-stimulated) radiation insertions. The experimental measurements were conducted at relatively high dose rates. It would have been interesting if the authors had examined dose rate as a variable in the induction of S-RI given that the preclinical CT dose response relationship showed a plateau while the higher dose rate Cs-137 irradiator did not. The results challenge conventional thinking regarding low dose effects and suggest that further study will be important to understand the implications of the work regarding the risks of radiation-induced detriment following medical imaging. As the authors acknowledge these observations would need to be demonstrated in a pre-clinical intact animal model, a much more difficult proposition, to impact current understanding of low dose effects. It is particularly important that the authors emphasize this to avoid having this work be misconstrued by other scientists or the general public.

**Have all data underlying the figures and results presented in the manuscript been provided?**

Reviewer #1: Yes

Reviewer #2: No: See point 3 in critique to authors; showing only normalized data makes it impossible to fully interpret it

Reviewer #3: Yes

PLOS authors have the option to publish the peer review history of their article (what does this mean?). If published, this will include your full peer review and any attached files.

Reviewer #1: No

Reviewer #2: No

Reviewer #3: Yes: George Sgouros

---

## [Editor Report · Decision Letter 1]

2 Dec 2019

Dear Roland,

We are pleased to inform you that your manuscript entitled "Low Dose Ionizing Radiation Strongly Stimulates Insertional Mutagenesis in a γH2AX Dependent Manner" has been editorially accepted for publication in PLOS Genetics. Congratulations!

Yours sincerely,

Nancy Maizels, Ph.D.

Associate Editor

PLOS Genetics

Gregory Barsh

Editor-in-Chief

PLOS Genetics

Comments from the reviewers (if applicable):

**Data Deposition**

http://datadryad.org/submit?journalID=pgenetics&manu=PGENETICS-D-19-01201R1

**Press Queries**

---

## [Editor Report · Acceptance letter]

18 Dec 2019

PGENETICS-D-19-01201R1 

Low Dose Ionizing Radiation Strongly Stimulates Insertional Mutagenesis in a γH2AX Dependent Manner 

Dear Dr Kanaar, 

We are pleased to inform you that your manuscript entitled "Low Dose Ionizing Radiation Strongly Stimulates Insertional Mutagenesis in a γH2AX Dependent Manner" has been formally accepted for publication in PLOS Genetics! Your manuscript is now with our production department and you will be notified of the publication date in due course.

With kind regards,

Kaitlin Butler

PLOS Genetics

On behalf of:
